# ACE-Vitamin Index and Risk of Glaucoma: The SUN Project

**DOI:** 10.3390/nu14235129

**Published:** 2022-12-02

**Authors:** Javier Moreno-Montañés, Elsa Gándara, Laura Moreno-Galarraga, Maria S. Hershey, José Francisco López-Gil, Stefanos Kales, Maira Bes-Rastrollo, Miguel Ángel Martínez-González, Alejandro Fernandez-Montero

**Affiliations:** 1Department of Ophthalmology, Clínica Universidad de Navarra, 31008 Pamplona, Spain; 2Instituto de Investigación Sanitaria de Navarra (IdiSNA), 31008 Pamplona, Spain; 3Department of Pediatrics, Complejo Hospitalario de Navarra, Servicio Navarro de Salud, 31008 Pamplona, Spain; 4Department of Environmental Health, T.H. Chan School of Public Health, Harvard University, Boston, MA 02138, USA; 5Health and Social Research Center, University of Castilla-La Mancha, 16071 Cuenca, Spain; 6Occupational Medicine, Cambridge Health Alliance, Harvard Medical School, Cambridge, MA 02115, USA; 7Department of Preventive Medicine and Public Health, School of Medicine, University of Navarra, 31008 Pamplona, Spain; 8CIBER Fisiopatología de la Obesidad y Nutrición (CIBER Obn), Instituto de Salud Carlos III, 28029 Madrid, Spain; 9Department of Nutrition, T.H. Chan School of Public Health, Harvard University, Boston, MA 02138, USA; 10Department of Occupational Medicine, University of Navarra, 31008 Pamplona, Spain

**Keywords:** vitamin A, vitamin C, vitamin E, glaucoma, SUN project

## Abstract

Background: Previous studies regarding antioxidant consumption and glaucoma have shown contradictory results. The aim of this study was to analyze the combined effect of the consumption of three vitamins (A, C and E) on the incidence of glaucoma in the SUN Project. Methods: For this study, 18,669 participants were included. The mean follow-up was 11.5 years. An index including vitamins A, C and E (ACE-Vitamin Index) was calculated. Vitamin intake was extracted from participants’ dietary data and vitamin supplements, if taken. Information on glaucoma incidence was collected by previously validated self-reported questionnaires. The association between glaucoma and vitamin intake was assessed by repeated-measures Cox regression using multi-adjusted hazard ratios. Results: A total of 251 (1.3%) cases of glaucoma were detected. Participants with a higher ACE-Vitamin Index presented a reduced risk of glaucoma compared to participants with lower consumption (adjusted HR = 0.73; 95% CI, (0.55–0.98)). When each vitamin was analyzed individually, none of them had a significant protective effect. The protective effect of the ACE-Vitamin Index was higher in men and older participants (≥55 year). Conclusions: The consumption of vitamins A, C and E considered separately do not seem to exert a protective effect against glaucoma, but when these vitamins are considered together, they are associated with a lower risk of glaucoma.

## 1. Introduction

Glaucoma is a frequent eye disease and includes a group of disorders characterized by progressive retinal ganglion cell (RGC) damage and optic nerve damage, associated with the loss of the visual field [1]. Glaucoma can slowly lead to blindness and is currently the second leading cause of irreversible blindness. It is therefore a public health concern, as it is estimated that by the year 2040, more than 110 million people will be affected by glaucoma [2]. The causes related to this eye disease are multifactorial, including aging and vascular, genetic, mechanical and immunological factors [3]. One hypothesis suggests that oxidative stress damages the aqueous humor drainage pathway at the level of the trabecular meshwork, which secondarily causes an increase in the intraocular pressure (IOP) and, as a consequence, damage to RGCs [4,5]. In addition, a low antioxidant capacity in the aqueous humor or at the systemic level has been associated with more advanced forms of glaucomatous damage and with more advanced visual field loss [5,6,7], so oxidative processes are proposed to play a role in glaucoma.

Oxidants and antioxidants constitute a diverse group of compounds with different properties. There are dietary components that may alter glaucoma risk. Among these, antioxidants are known to inhibit oxidative events, inhibit oxidation formation and repair oxidant-induced injury [8]. The antioxidant defense system includes both endogenous and exogenous compounds. In a balanced diet, there are multiple common nutrients in which we can find exogenous substances with direct and indirect antioxidant activity, such as vitamins A, C and E. Although there is no clear evidence that vitamin A itself has antioxidant effects, this vitamin acts indirectly as an antioxidant due to its effects on the transcription of genes necessary to provide an effective antioxidant response [9].

The role of these vitamins in glaucoma is controversial due to the conflicting results of studies conducted to date. However, there is robust evidence that oxidative stress is one of the causes of glaucoma pathogenesis. According to a recent meta-analysis [10], vitamins A and C might seem to play a protective role in glaucoma, although there is currently insufficient evidence demonstrating a link between serum vitamin levels (e.g., vitamins A and C) and glaucoma.

The *Seguimiento Universidad de Navarra* cohort (SUN Project) is an open, prospective, multidisciplinary cohort with more than 20,000 participants and a follow-up of more than 20 years. A previous study of this cohort indicated that high adherence to the Mediterranean lifestyle was a protective factor against glaucoma [11]. The objective of this study was to investigate whether vitamins A, C and E had a protective effect on glaucoma incidence. 

## 2. Materials and Methods

### 2.1. Study Population

The SUN Project is a large prospective cohort formed by a Mediterranean population of Spanish college graduates. It has been designed to identify dietary and lifestyle determinants of different diseases, such as cardiovascular diseases, mental diseases or cancer [12]. It started in 1999, and it is a dynamic open cohort, so recruitment is permanently open. Data collection and participant follow-up are performed using biennial questionnaires sent by mail or via the web. The SUN cohort profile can be found at: www.medpreventiva.es/xZd6Hh (accessed on 30 November 2022).

The first questionnaire (Q0 or baseline questionnaire) includes information on participants’ sociodemographic data, anthropometric data, diet and eating behaviors, as well as lifestyle habits and medical data. Every two years, follow-up questionnaires (Q2–Q18) evaluate changes in diet, lifestyle and medical data and determine the incidence of new diseases. After the first 10 years of follow-up, using the Q10 questionnaire, general data are collected again to further update the behaviors and lifestyles of the participants. Once participants understood the necessary information and the SUN Project method, informed consent was implied when answering Q0. Specific consent was obtained from participants before consulting their medical records or including them in validation studies. Candidates were informed of their right to refuse to participate and withdraw their consent to participate at any time, in accordance with the Declaration of Helsinki. The University of Navarra Institutional Review Board approved the SUN Project (091/2008).

By September 2017, 22,553 participants had been recruited for the SUN Project. Participants with prevalent glaucoma on the baseline questionnaire and other exclusion factors are shown in a flow chart in Figure 1. A total of 18,669 participants were included in the analysis, with a mean follow-up period of 11.5 years (SD: 4.5). 

### 2.2. Assessment of ACE-Vitamin Index

Three vitamins with direct or indirect antioxidative functions were included in this study. The total intake of each vitamin was calculated following local food composition tables and included intake from all dietary sources available (Table 1) [14]. Dietary vitamin consumption was assessed through a 136-item semi-quantitative food consumption questionnaire that was previously validated [15], as well as from specific vitamin supplements (if taken). Both the baseline questionnaire (Q0) and Q10 questionnaire were considered. To calculate the ACE-Vitamin Index, one point was scored for each of the three vitamins (A, C and E) for participants with a vitamin consumption above the median consumption of the SUN population. Thus, participants who were above the median were given one point, and those who were below or equal to the median consumption were given no points. As three vitamins were studied, the index ranged from zero to three points. Vitamin consumption was classified as high (if the total sum of the ACE-Vitamin Index was two or three points) or low (if the total sum was zero or one point).

### 2.3. Outcome Assessment: Glaucoma Incidence

Glaucoma diagnosis was assessed on Q0 and biannually on each of the follow-up questionnaires. Prevalent cases of glaucoma were not included in the analysis. Participants were asked the following question: “Have you ever been diagnosed with glaucoma by a health care professional?” The date of diagnosis was also requested. The participants’ self-reported diagnosis of glaucoma was clinically validated by an ophthalmologist in a subsample of 150 participants, comparing the self-reported diagnosis to the clinical diagnosis. Glaucoma was defined, following the European Glaucoma Society recommendations, as a visual field defect with damage to the optic nerve rim and loss of retinal nerve fibers. A correct glaucoma diagnosis was found in the validation study performed, with a Kappa value of 0.85 (95% coefficient interval (CI), 0.83-0.87) and good sensitivity (0.83) and specificity (0.99) [16].

### 2.4. Assessment of Other Covariates

Information on several potential confounding factors was assessed at baseline. Covariates such as socio-demographic data (sex, age and educational level), lifestyle- and health-related characteristics (smoking, physical activity, total energy intake, special diet consumption, caffeine intake and omega-3/6 ratio), anthropometric measures (e.g., body mass index) and personal medical history (prevalent diseases such as hypertension, cardiovascular disease, cancer or diabetes).

### 2.5. Statistical Analysis

For repeated measures, Cox regression models were used. To analyze the glaucoma risk according to participants’ vitamin consumption, the low-vitamin-intake group (total ACE-Vitamin Index sum of 0 or 1) was used as the reference category versus high vitamin intake (total sum of 2 or 3). Analyses compared the incidence of glaucoma between each group. The HRs and their 95% confidence intervals were used as measures of effect. In models with successive degrees of adjustment, crude HRs were estimated, as well as HRs adjusted for sex and age and multivariable-adjusted HRs for several potential confounders, such as age, sex, educational level, body mass index in quintiles, smoking measured as pack-years in quintiles, total energy intake presented in quintiles, leisure-time physical activity (METS-h/week) in quintiles, caffeine consumption in quintiles, alcohol consumption in quintiles, the omega-3/omega-6 ratio in quintiles, cancer, hypertension, diabetes, cardiovascular disease and whether they had been on any special diets. An additional adjustment was also performed to include the consumption of polyphenols and flavonoids derived from participants’ food intake. These same analyses were performed depending on whether the source of vitamin intake was from the participant’s diet or from vitamin supplements.

To analyze the contribution of each vitamin in the index, Cox regression models were fitted for the three different vitamins, and a multivariable adjustment was performed using the same variables as in the previous analysis and considering the participants in the lower-vitamin-intake category (below or equal to the median) as the reference category. To ensure the robustness of the results, a multi-adjusted Cox regression model was also performed using an extended index by adding carotenoid intake to the ACE-Vitamin Index and then dichotomizing the new extended score into low nutrient intake (0–1) and high nutrient intake (2–4). Stratified analyses and tests for interactions were also performed with sex and age (<50 years and ≥50 years). All *p*-values are two-tailed, and *p* < 0.05 was considered statistically significant. Analyses were performed with STATA 12.0 software.

## 3. Results

In this study, 18,669 participants were included in the analysis; 60% were women, and the mean age was 38.3 (SD 12.3). During the follow-up (mean follow-up time: 11.5 years, SD: 4.5), 251 new incident cases of glaucoma were identified over a total of 193,737 person-years. 

The distribution of baseline characteristics according to participants’ ACE-Vitamin Index is summarized in Table 2. Participants with higher scores were less likely to be men, smoked less, had higher energy intake, practiced more physical activity and were more likely to have DM type II.

Table 3 shows the risk of glaucoma according to the ACE-Vitamin Index. Higher vitamin consumption was inversely associated with the risk of glaucoma, showing a multivariable-adjusted HR of 0.73 (95% CI (0.55–0.98)).

The daily amount of vitamin intake (median and interquartile range) that added a point to the ACE-Vitamin Index was more than 2530.7 ugr/day (2086.4–3442.3) for vitamin A, 357.1 mgr/day (299.1–448.2) for vitamin C and 8.85 mgr/day (7.28–11.70) for vitamin E. Figure 2 shows the risk of glaucoma for each vitamin: HR for vitamin A: 0.79, 95% CI (0.60–1.04); HR for vitamin C: 0.94, 95% CI (0.71–1.44); and HR for vitamin E: 1.13, 95% CI (0.84–1.52). When each vitamin was analyzed individually, none of them had a significant protective effect (Figure 2).

When the source of each vitamin’s intake was analyzed in our population, it was observed that supplements contributed only 3.4%, 1.5% and 8.6% of the total intake of vitamins A, C and E, respectively. Food was the main source of the intake of vitamins, showing a multivariable-adjusted HR in the highest-vitamin-intake group of 0.73 (95% CI (0.54–0.97)). 

In the subgroup analysis (Figure 3), after stratification by sex and age, a significant multiplicative interaction was not found, although there was a stronger inverse association between the ACE-Vitamin Index and glaucoma in men and in older participants (older than 50 years old).

Finally, when carotenoid intake was added to the extended vitamin index (Vitamin A, C, E, and carotenoids) a similar inverse association with greater magnitude and significance HR 0.67, 95% CI (0.50 to 0.89), was observed in the group with the highest score compared to the group with the lower score.

## 4. Discussion

In this study, the relationship between each vitamin’s consumption (vitamin A, C or E) and glaucoma showed no significant association when analyzed independently. Conversely, when analyzed together (i.e., ACE-Vitamin Index), the combination of these three vitamins was inversely associated with the risk of developing glaucoma. 

Previous studies have suggested that nutrition may influence vascularization, IOP and optic nerve damage in glaucoma [8,17]; however, the effect of antioxidant vitamins remains controversial. It is well known that one of the causes of glaucoma is oxidative stress, as it causes damage to the trabecular meshwork, resulting in increased IOP and, if prolonged over time, could lead to loss of RGCs and optic nerve fibers [4,5,6,7].

Vitamin A can be obtained as preformed vitamin A (such as retinol) and provitamin A (such as beta-carotene). It is a fat-soluble vitamin that can be stored in the liver, and it is therefore unnecessary to take it daily [18]. The Recommended Dietary Allowance for adults is 900 μg daily for men and 700 μg for women [19], and our participants showed a median vitamin A consumption of 2530 μg/day. Although there is no clear evidence that vitamin A itself has antioxidant effects, this vitamin can act indirectly as an antioxidant due to its effect on the transcription of certain genes that are necessary for the body to provide an effective antioxidant response [9]. Therefore, in this study, this vitamin was also included in the vitamin index. Vitamin A is related to white blood cell function, bone remodeling and endothelial cell maintenance, and it has a favorable immune-modulating effect [20,21], but one of its most important functions is its role in visual phototransduction, facilitating the process of the transformation of light photons into electrical signals. It also serves to protect the conjunctiva and to alleviate symptoms of dry eye [18]. Several studies have previously analyzed the relationship between this vitamin and the incidence of glaucoma, most of which did not find any significant relationship [22]. Only two studies regarding the dietary intake of retinol equivalents found a protective effect on open-angle glaucoma [23,24]. Our cohort study, as in other big-cohort studies, such as the Health Professionals Follow-up Study and the Nurses’ Health Study with more than 100,000 participants, did not find a significant association between vitamin A and glaucoma incidence when analyzed individually [25].

Vitamin C, also known as ascorbic acid, is a water-soluble vitamin and is important for its immunological properties [26], its tissue regeneration effect due to collagen formation and its antioxidant properties. It is also essential for the primary prevention of various diseases, such as coronary heart disease, stroke, infections and cancer [27]. The Recommended Dietary Allowance for adults is 90 mg daily for men and 75 mg for women [19], while the median vitamin C consumption among our participants was 357 mg/day. A higher intake of vitamin C has been linked to an increased concentration in the aqueous humor [28]. Regarding of effects of glaucoma on eye diseases, an association between patients with normal-tension glaucoma and lower serum levels of vitamin C has been published [29]. Additionally, a study with African American participants from the Osteoporotic Fractures Study showed that higher vitamin C intake was associated with a lower risk of glaucoma [23]. Finally, a study including 2912 participants in the National Health and Nutrition Examination Survey (NHANES) found that the consumption of high-dose vitamin C supplements was associated with decreased odds of glaucoma, although serum vitamin C levels did not correlate with the glaucoma prevalence [22]. It is important to highlight that these results may be biased by the fact that an important source of vitamin C is leafy green vegetables, and several studies have provided evidence of an association between leafy green vegetables and glaucoma due to the protective effect of the nitrates present in these vegetables [30,31]. In line with our results, other studies and meta-analyses did not find an association between vitamin C and glaucoma [10,23,24,25].

Vitamin E is a fat-soluble antioxidant vitamin with several forms, including tocopherols (as alpha-tocopherol) and tocotrienols. The Recommended Dietary Allowance for adults is 15 mg daily for men and women [19]; however, we found a median vitamin E consumption of 8.85 mg/day in our participants. Vitamin E is an essential micronutrient that participates in the prevention and treatment of cardiovascular, neurological and aging-related diseases [32,33] and also decreases insulin resistance and other cardiometabolic risk factors [34]. Regarding eye health, low levels of vitamin E in the aqueous humor have been associated with glaucoma [35], while other studies on dietary vitamin E intake and OAG did not reveal significant associations [22,23,24]. Supporting our findings, a meta-analysis found no association between glaucoma and vitamin E intake [10].

On the other hand, we found that a higher ACE-Vitamin Index was associated with a lower risk of glaucoma (multivariable-adjusted HR of 0.73, 95% CI (0.55–0.98)). The protective effect against glaucoma may be due to a possible synergic antioxidant effect of these three vitamins. Previous studies have mainly focused on the study of each vitamin individually, but to be able to determine their effects, they should be further analyzed as a combination by creating indexes or scores including different exogenous antioxidant combinations. 

The biological pathway of this association has been previously studied. Some studies indicated that oxidative stress can play an important role in both the induction and maintenance of optic nerve degeneration and RGCs as part of the neurodegenerative process [8,35,36,37,38]. Although the exact mechanism by which RGC death occurs in glaucoma is unknown, it has been observed that these cells are particularly vulnerable to raised levels of oxidative stress due to their high oxygen consumption and their high proportion of polyunsaturated fatty acids. In the two theories used to explain the pathophysiology of glaucoma, vascular and mechanical, RGC death is mediated by oxidative stress [8]. Likewise, oxidative stress also causes damage to the trabecular meshwork that alters the outflow of aqueous humor and, therefore, increases IOP with secondary damage to RGCs [8,37,38]. Therefore, oxidative stress is involved in both IOP-dependent and -independent mechanisms.

Other ocular pathologies also improve if vitamins or antioxidants are consumed. An example of this is age-related macular degeneration (AMD). The recommendation of high doses of antioxidant foods/supplements is mainly based on two highly relevant studies, known as AREDS 1 (Age-Related Eye Disease Study) and AREDS 2 [39,40]. According to AREDS, the intake of a combination of antioxidants (beta-carotene and vitamins A, C and E) together with zinc was associated with a reduction in the progression of AMD compared to placebo (OR: 0.72, 99% CI: 0.52–0.98) [39]. The AREDS 2 study recommended replacing beta-carotene with a combination of lutein and zeaxanthin, given the increased incidence of lung cancer [40]. Currently, supplementation with combinations such as those used in the AREDS study is recommended in patients with AMD to prevent disease progression. 

In this study, the intake of vitamins included vitamins from both the diet and supplements. However, for the three vitamins studied, the principal source was the diet. The results showed that the inverse association persists when only vitamins from the diet were analyzed. It should be considered that this is a Mediterranean population, and with a Mediterranean diet, rich in antioxidant foods (e.g., fruits and vegetables), there is a sufficient concentration of antioxidant vitamins. The contribution to the total intake of the three vitamins from vitamin supplements was low (3.4%, 1.5% and 8.6% of the total intake of vitamins A, C and E, respectively). This finding may suggest that a varied and balanced diet may be an appropriate way to obtain the health benefits of vitamins.

In additional sensitivity analyses, we added carotenoids (vitamin A precursors with direct antioxidant functions) to the ACE-Vitamin Index, generating an extended vitamin index (vitamins A, C and E and carotenoids), and we could see that the protective association persisted and that the magnitude of the HR became more pronounced. We also performed a sensitivity analysis including an extra adjustment for other possible sources of antioxidants from the diet. After adjusting for the consumption of polyphenols and flavonoids, the index formed by these vitamins continued to protect against glaucoma. We believe these findings could reinforce the oxidative theory of the risk of glaucoma. When we add an antioxidant to our index, the protective effect is greater since the antioxidant action is strengthened, but more in-depth studies are needed to better understand the pathways and biological mechanisms behind these findings, and further studies should include possible cofounders; for example, vitamin absorption, endogenous antioxidants and other substances that can affect the final oxidative balance need to be better addressed. In the subanalysis stratified by age and sex, a greater protective effect of the ACE-Vitamin Index against glaucoma was seen in male participants and in older participants. It appears that these higher-risk groups may have more benefits from vitamin intake than women and young people. These observed differences might be explained by the lower endogenous antioxidant compounds in these groups. Another possible explanation is that, as glaucoma is a disease of older people, we have not followed the population long enough to see enough cases of glaucoma in the younger population.

This study has several strengths and limitations that need to be addressed. Since the SUN cohort is mainly composed of healthy middle-aged college graduates, the incidence of glaucoma in our sample is low, given that glaucoma increases exponentially with age (0.07% at age 40 and 6.9% at age 80), but the large sample size allowed us to perform analyses with sufficient statistical power. Secondly, it is important to note that each food contains hundreds of chemical components that tend to interrelate, making it difficult to interpret the possible relationship with the disease. In fact, nutritional epidemiology initially analyzed the effect of nutrients individually from a biochemical point of view, but the classical analytical approach has increasingly shifted toward a greater interest in the study of overall dietary patterns. This shift is due in part to the possible synergistic or antagonistic effects of consuming portions of foods in combination, since we do not consume nutrients but rather foods within a dietary pattern [41,42]. In addition, there are differences in dietary patterns and habits, such as the cleaning, preparation and storage of the same food, that can modify the absorption and metabolism of nutrients. Actions such as chopping and cooking vegetables in oil often increase nutrients’ bioavailability [41,42]. In our study, we could not differentiate these aspects in the vitamin intake calculation. Despite these limitations, there are strengths we would like to point out, including the high number of participants in the cohort, as well as the long follow-up period, the use of repeated measures and the high retention rate (91.1%).

In conclusion, this prospective cohort study conducted in a Mediterranean population shows that participants with a higher ACE-Vitamin Index (combination of vitamins A, C and E) have a lower risk of developing glaucoma when compared to participants with a lower ACE-Vitamin score, possibly due to a synergistic antioxidant effect between these vitamins. This protective effect is found with the consumption of vitamins solely from the diet (no supplements are needed) and is stronger among males and older participants. Further studies are needed to assess whether the same results are obtained in other populations and to establish the best dietary recommendation for the prevention of glaucoma.

## Figures and Tables

**Figure 1 nutrients-14-05129-f001:**
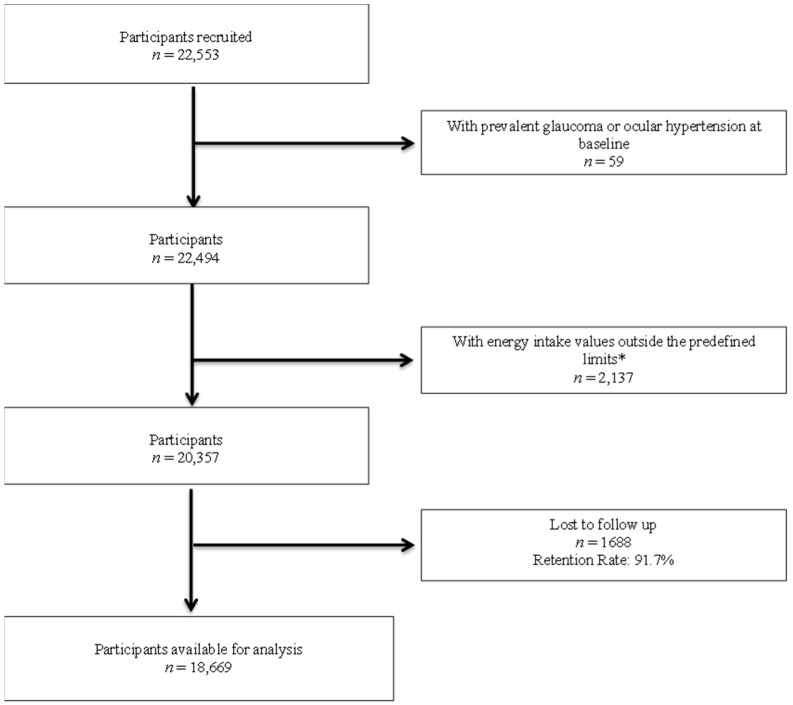
Flow chart of participants in the *Seguimiento Universidad de Navarra* (SUN) Project. * Values outside of predefined limits according to Willett [13].

**Figure 2 nutrients-14-05129-f002:**
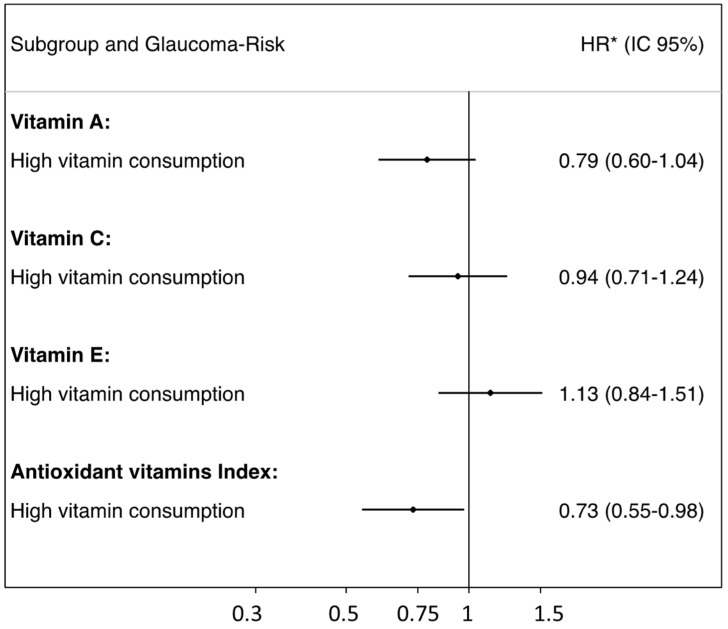
Glaucoma risk according to each vitamin’s consumption. The SUN cohort. * Adjusted for body mass index, baseline cancer prevalence, arterial hypertension, cigarette and caffeine consumption, physical exercise, omega-3/6 ratio, type 2 diabetes mellitus, baseline educational level and year in which the baseline questionnaire was completed.

**Figure 3 nutrients-14-05129-f003:**
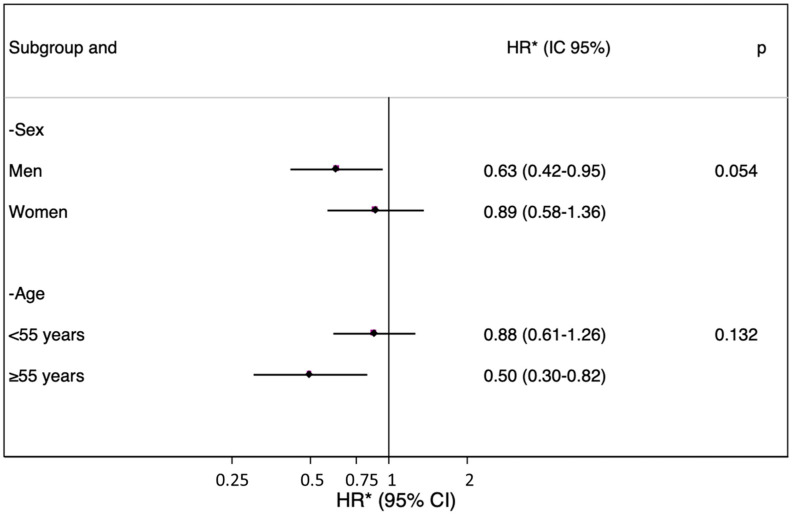
Stratified analysis by sex and age: Glaucoma risk in participants with high ACE-Vitamin Index vs. low ACE-Vitamin Index (category reference). The SUN cohort. HR: hazard ratio. * Adjusted for body mass index, baseline cancer prevalence, arterial hypertension, cigarette and caffeine consumption, physical exercise, omega-3/6 ratio, type 2 diabetes mellitus, baseline educational level and year in which the baseline questionnaire was completed.

**Table 1 nutrients-14-05129-t001:** Vitamins analyzed and foods considered as vitamin sources for each vitamin.

Vitamins	Included Foods
Vitamin A	Whole milk, condensed milk, cream, milkshakes, yogurt, petit suisse, curds, portioned or cream cheese, cured or semi-cured cheese, fresh cheese, custard, eggs, liver, pate, white fish, blue fish, cod, salted and/or smoked fish, mussels, oysters, prawns, shrimps, squids, octopus, chard, cabbage, lettuce, tomato, carrot, beans, eggplant, orange, banana, apple, strawberries, peach, cherries, figs, watermelon, melon, fruits in syrup, dried fruits, dried fruits, olives, avocado, kiwi, lentils, chickpeas, peas, pizza, butter, margarine, donuts, industrial pastries, homemade pastries, cakes, pastries, natural juices, bottled juices, croquettes, mayonnaise, tomato sauce and jam.
Vitamin C	Whole milk, semi-skimmed or skimmed milk, custard, milkshakes, yogurt, chicken with skin or without skin, liver, viscera, chard, cabbage, lettuce, tomato, carrot, beans, eggplant, peppers, asparagus, green vegetables, French fries, baked potatoes, orange, banana, apple, strawberries, peaches, cherries, figs, watermelon, melon, grapes, fruits in syrup, fruits in juice, avocado, mango, kiwi, lentils, chickpeas, peas, cereals, pizza, donuts, natural orange juice, other natural juices, croquettes, tomato sauce and jam.
Vitamin E	Whole milk, semi-skimmed milk, condensed milk, whipped custard, yogurt, curd, cheese in portions or cream cheese, cured or semi-cured cheese, fresh cheese, eggs, chicken with or without skin, veal, pork, lamb, rabbit, liver, other offal, cured ham, sausages, sausages, blood sausage, hamburger, bacon, blue fish, cod, salted or smoked fish, mussels, oysters, prawns, shrimps, squids, chard, beans, eggplants, peppers, French fries, baked potatoes, bananas, nuts, dried fruits, avocado, lentils, chickpeas, beans, peas, black bread, rice, pizza, butter, margarine, olive oil, sunflower oil, corn oil, cookies, muffins, donuts, churros, chocolate, chocolates, chocolates, nougat, marzipan, crop nuts, soups and creams, tomato sauce and mayonnaise.

**Table 2 nutrients-14-05129-t002:** Baseline characteristics of participants by ACE-vitamin intake. Values are expressed as means (SD), unless otherwise noted (SUN Project).

Participants’ Baseline Characteristics	Low Vitamin Consumption	High Vitamin Consumption
ACE-Vitamin Index	0–1 point	2–3 points
*n*	9343	9326
Age, years	38.1 (12.2)	38.6 (12.4)
Sex. men, %	46.79	33.14
Educational level, %		
College	21.73	25.86
Postgraduate	50.30	46.96
Master	8.20	7.91
Doctorate	10.53	9.76
Missing or no college	9.24	9.51
Body mass index, kg/m^2^	23.8 (3.6)	23.3 (3.5)
Physical activity, METS-h/week	19.4 (20.3)	24.1 (25.0)
Total energy intake, Kcal/day	2119.5 (582.5)	2568.1 (568.4)
Omega 3:6 ratio	0.2 (0.1)	0.2 (0.1)
Hypertension, %	11.07	10.46
Cancer, %	2.30	2.81
Type 2 diabetes, %	1.56	2.34
Smoking, packages/year	6.01(10.9)	4.8 (9.1)
Alcohol intake, mg/day	7.1 (11.2)	6.2 (9.1)
Caffeine intake, mg/day	42.5 (40.4)	42.3 (39.0)
Special diet, %	6.75	10.07

**Table 3 nutrients-14-05129-t003:** Glaucoma risk according to ACE-vitamin consumption. The SUN cohort.

	Low Vitamin Consumption	High Vitamin Consumption
ACE-Vitamin Index	0–1 point	2–3 points
Persons/year	97,252	95,148
New glaucoma cases, *n*	120	146
Crude hazard ratio HR (95% CI)	1 (ref.)	0.91 (0.71–1.18)
Sex- and age-adjusted HR (95% CI)	1 (ref.)	0.87 (0.67–0.76)
Multivariable-adjusted HR (95% CI) *	1 (ref.)	0.73 (0.55–0.98)
Multivariable-adjusted HR (95% CI) **	1 (ref.)	0.72 (0.52–0.99)

HR: hazard ratio. Ref.: reference. * Adjusted for body mass index, baseline cancer prevalence, arterial hypertension, cigarette and caffeine consumption, physical exercise, omega-3/6 ratio, type 2 diabetes mellitus, baseline educational level and year in which the baseline questionnaire was completed. ** Multivariable-adjusted and including polyphenol and flavonoid intake.

## Data Availability

Available upon request from the Department of Preventive Medicine and Public Health, School of Medicine, University of Navarra, Pamplona, Spain.

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
