# Peer review of "ACE-Vitamin Index and Risk of Glaucoma: The SUN Project"

_nutrients, 2022, doi:10.3390/nu14235129_

Round 1
Reviewer 1 Report
Overall Comments:
1. Having been involved in vitamin A, carotenoid, vitamin E and vitamin C research for almost 50 years, I have familiarity with the accurate terminology regarding these compounds and their physiological effects. More specifically, for the authors to label vitamin A as an antioxidant vitamin is problematic. Although some of the carotenoid precursors for vitamin A are antioxidants, there is no evidence that vitamin A, itself, has antioxidant capacity. I suggest the authors read the recent review article by Bill Blaner, who has similarly done research in vitamin A, (Ann. Rev. Nutrition, 41:105-131, 2021) I wholeheartedly agree with his conclusion that vitamin A is NOT an antioxidant.
I do realize that the Spanish diet has great similarity with the Mediterranean Diet and thus individuals from Spain would be anticipated to have a large fruit and vegetable intake, but the data on the subjects does not indicate the fraction of carotenoids nor actual vitamin A being consumed. This is a serious flaw and will necessitate a re-evaluation of the dietary data and a re-writing of the manuscript to reflect the contribution of the carotenoid fraction, the vitamin A component, as well as the combinatorial value of these components with the vitamin E and vitamin C intake.
2. The extraction of intake of vitamins from the food consumption questionnaire needs to be discussed in detail so that the data obtained has context.
SPECIFIC COMMENTS:
l. 110; change diagnose to diagnosis
l. 174-75; the sentence is awkwardly expressed and needs to be rewritten.
Author Response
Thanks for your review. Your comments help us and encourage us to improve our research.We request your questions point by point below. The corrections suggested by you and other reviewers are highlighted in the original manuscript in red print.
- Having been involved in vitamin A, carotenoid, vitamin E and vitamin C research for almost 50 years, I have familiarity with the accurate terminology regarding these compounds and their physiological effects. More specifically, for the authors to label vitamin A as an antioxidant vitamin is problematic. Although some of the carotenoid precursors for vitamin A are antioxidants, there is no evidence that vitamin A, itself, has antioxidant capacity. I suggest the authors read the recent review article by Bill Blaner, who has similarly done research in vitamin A, (Ann. Rev. Nutrition, 41:105-131, 2021) I wholeheartedly agree with his conclusion that vitamin A is NOT an antioxidant.
I do realize that the Spanish diet has great similarity with the Mediterranean Diet and thus individuals from Spain would be anticipated to have a large fruit and vegetable intake, but the data on the subjects does not indicate the fraction of carotenoids nor actual vitamin A being consumed. This is a serious flaw and will necessitate a re-evaluation of the dietary data and a re-writing of the manuscript to reflect the contribution of the carotenoid fraction, the vitamin A component, as well as the combinatorial value of these components with the vitamin E and vitamin C intake.
Thank you very much for your suggestions and comments. It is great to have someone with such a wide experience on the topic, and we believe the paper has improved with your suggestions.
The recommended article from Blaner, has been a very interesting read, and we agree with all your comments regarding vitamins. Reading the article, it is clear that vitamin A acts as an indirect antioxidant, implicated in the transcription of genes involved in mediating the body´s antioxidant responses. We have added the reference in the new version of the manuscript and we have added specific comments on vitamin A, explaining this.
The next sentence has been added, and changes have been made: "Although there is no clear evidence that vitamin A, itself; has antioxidant effects, this vitamin can act indirectly as an antioxidant, having effect on the transcription of certain genes that are necessary for the body to provide an effective antioxidant response”. This is why for the study this vitamin has also been included in the same vitamin index.
Following your suggestions, we have actually changed the title, using the term “ACE vitamins” instead of “ANTIOXIDANT vitamins”, and we have explained this in the methods section.
Regarding the use of carotenoids, as you suggested, we have performed a new sub analysis adding the carotenoids intake into the vitamin score. The results obtained show an inverse association of greater magnitude and significance HR 0.67; CI95% (0.50 to 0.89) when we compared the group of participants with the highest intake of these nutrients (2-4) versus the lowest intake (0-1) in this new nutrient score. We have added this analysis to the new version of the manuscript. Thanks
- The extraction of intake of vitamins from the food consumption questionnaire needs to be discussed in detail so that the data obtained has context.
Thank you for your comment. We have better explained how the extraction of vitamin intake was performed. We used a 136 items FFQ, already validated. We have added a table with the foods consumed in our population that provide for each of the vitamins analyzed. We have also included the bibliographic references of the methods and tables used for the calculation.
SPECIFIC COMMENTS:
- 110; change diagnose to diagnosis.
Done
- 174-75; the sentence is awkwardly expressed and needs to be rewritten.
Done.
Reviewer 2 Report
Comments:
Q1: Abstract: As stated, “Previous studies regarding antioxidant consumption and glaucoma have shown inconclusive results”, does it imply that this article will make definite results? “When evaluated separately, none of the three vitamins was significantly associated with glaucoma.” Thus, I would suggest to improve the relevant expressions. “This protective effect was higher in men and older participants”, whether there is a significant difference observed according to this sentence?
Q2: Introduction: “there is robust evidence that oxidative stress is one of the causes of glaucoma pathogenesis”, then the link between serum vitamin levels (e.g., vitamin A, C, E) and glaucoma, and the controversial conclusion were suggested to be further expounded, and the related functions of vitamin A, C, E were recommended to be supplemented in this section. Antioxidant active substances in foods were not the only vitamins, did the authors consider the effects of compound intake like polyphenols, flavonoids on the results presented here.
Q3: Materials and Methods: The Assessment of Antioxidant Vitamin Index section was suggested to be improved. How could the authors accurately calculate the amount of various vitamins in various foods for each person, and the definition of high and low vitamin consumption was confused.
Q4: Results: I did not find the expression of p < 0.05, thus whether the difference was significant could not be intuitively judged. In Fig. 2, the results were interesting, what is the relationship of Antioxidant Vitamin Index and the sum of vitamins A, C, and E. According to these results, did vitamin A intake present more crucial effects on the glaucoma risk?
Q5: Discussion: In this section, the Recommended Dietary Allowance of vitamins A, C, and E for adults were quite different from participants’ intakes, did these dosages negatively affect the results analysis and the health of participants? Besides, I also considered that various bioactive compounds in foods could exhibit good antioxidant effects, did these components interfere the authors analysis, if authors only consider the relationship between antioxidant and glaucoma? Whether the absorption of vitamins will also affect the results presented here, and whether the authors have considered it? If possible, I would suggest the authors supplement the reasons of stronger inverse association between the Antioxidant Vitamin Index and glaucoma in men and in older participants (older than 50 years-old).
Q6: Conclusions: These findings were interesting, but the depth of the analysis still needs to be further improved.
Q7: References: The formats of these references needs to be further modified, and the references in the last three years need to be further supplemented.
Author Response
Thanks for your review. Your comments help us and encourage us to improve our research.We request your questions point by point below. The corrections suggested by you and other reviewers are highlighted in the original manuscript in red print.
Q1: Abstract: As stated, “Previous studies regarding antioxidant consumption and glaucoma have shown inconclusive results”, does it imply that this article will make definite results? “When evaluated separately, none of the three vitamins was significantly associated with glaucoma.” Thus, I would suggest to improve the relevant expressions. “This protective effect was higher in men and older participants”, whether there is a significant difference observed according to this sentence?
Thanks for your comments and your suggestions. We have adapted the abstract.
The following sentence has been changed: Previous studies regarding antioxidant consumption and glaucoma have shown contradictory results. To avoid using inconclusive. Also, we have clarified the other mentioned expressions on the abstract.
Regarding the sentence: “This protective effect was higher in men and older participants” we performed stratified analyses, that can be seen in figure 2 and we found significant differences in the males and in the older age group participants, but not in the females nor in the younger age group participants. We have included a possible explanation for these findings in the discussion session. Thanks.
Q2: Introduction: “there is robust evidence that oxidative stress is one of the causes of glaucoma pathogenesis”, then the link between serum vitamin levels (e.g., vitamin A, C, E) and glaucoma, and the controversial conclusion were suggested to be further expounded, and the related functions of vitamin A, C, E were recommended to be supplemented in this section. Antioxidant active substances in foods were not the only vitamins, did the authors consider the effects of compound intake like polyphenols, flavonoids on the results presented here.
Thanks again for your suggestion. We had not considered them before, but after reading your suggestions, we also think they should be considered. We have made a new additional adjustment adding these components in the regression analyses (new Table 3) to better assess the effect of the ACE-Vitamin index on glaucoma independently of these other antioxidant substances.
Q3: Materials and Methods: The Assessment of Antioxidant Vitamin Index section was suggested to be improved. How could the authors accurately calculate the amount of various vitamins in various foods for each person, and the definition of high and low vitamin consumption was confused.
Thank you for the suggestion, we agree, this point needed to be improved as several reviewers were concerned about it. To improve comprehension, a new table has been added with all the foods that were taken into account for the calculation of the consumption of each vitamin, also the methods and tables used have been added, including the references of the specific tables that were used, to estimate the composition of each food.
Q4: Results: I did not find the expression of p < 0.05, thus whether the difference was significant could not be intuitively judged. In Fig. 2, the results were interesting, what is the relationship of Antioxidant Vitamin Index and the sum of vitamins A, C, and E. According to these results, did vitamin A intake present more crucial effects on the glaucoma risk?
Thanks, we have better explained. When using 95% Confidence Intervals in a cox regression, if the null value (1) is not included, p value is <0.05 and the expression p<0.05 can be found at the end of the statistical analysis section.
The sentence explaining Figure 2 has been rewritten in the results to make it more understandable. “When each vitamin was analyzed individually, none of them had a significant protective effect” but as seen in results, the vitamin that showed a greater protective effect was vitamin A. This has also been better addressed in the discussion section.
Q5: Discussion: In this section, the Recommended Dietary Allowance of vitamins A, C, and E for adults were quite different from participants’ intakes, did these dosages negatively affect the results analysis and the health of participants? Besides, I also considered that various bioactive compounds in foods could exhibit good antioxidant effects, did these components interfere the authors analysis, if authors only consider the relationship between antioxidant and glaucoma? Whether the absorption of vitamins will also affect the results presented here, and whether the authors have considered it? If possible, I would suggest the authors supplement the reasons of stronger inverse association between the Antioxidant Vitamin Index and glaucoma in men and in older participants (older than 50 years-old).
Thanks for the suggestions, we have tried to address all your concerns:
In the new additional sensitivity analyses performed, we found that when carotenoid intake (a precursor to vitamin A with a direct antioxidant function) is added to the vitamin index the magnitude of the HR remains similar, becoming slightly more pronounced. We also made new additional adjustment in the ACE-Vitamine index adding other antioxidants ( as polyphenos and flavonoids) in the regression analyses (new Table 3) also getting similar HR values. Therefore, ACE-Vitamin index is associated with glaucoma independently of these other antioxidant substances.
We believe these findings could reinforce the oxidative theory and the risk of glaucoma. When we add antioxidant to our index the protective effect is greater, since as the antioxidant action is strengthened in the new index. But it is also interesting to note that when adjusting by the consumption of polyphenols and flavonoids, the index formed by these 3 vitamins continues to protect against the risk of glaucoma. We have added new information on the discussion. Deeper studies are needed to better understand the pathways and biological mechanism behind these findings, further studies including possible cofounders such as vitamin absorption, as you mention, and other substances that can affect the oxidative balance need to be addressed.
We have also added the following sentence, regarding your comment, to explain a possible hypothesis about the effect we observed in the older participants. "This difference could be explained by the lower endogenous antioxidant compounds in older participants. Another possible explanation is that, as glaucoma is a disease of older people, we have not followed the population long enough to see enough cases of glaucoma appear in the younger population.
Q6: Conclusions: These findings were interesting, but the depth of the analysis still needs to be further improved.
Thanks for your kind comment. We hope in this new version, and with the new sensitivity analysis performed we have improved our manuscript thanks to the addition of the reviewers comments.
Q7: References: The formats of these references needs to be further modified, and the references in the last three years need to be further supplemented.
Thank you, new more current references have been added.
Reviewer 3 Report
Thank you for the opportunity to review this manuscript.
The ubiquitous exposure to vitamins has led to many epidemiological studies for many diseases. Glaucoma is no exception. Several studies suggested that nutrition might affect intraocular pressure or glaucoma mediated by oxidative stress. Oxidative stress occurs when more reactive oxygen species are formed than the anti-oxidative capacity of the cell can handle.
This study aims to investigate whether high intakes of some antioxidant vitamins, such as vitamins A, C, and E, have a protective effect on glaucoma.
I have some suggestions to improve this manuscript. In particular, the introduction should better investigate the role of vitamins analyzed in clinical management and proposal in this manuscript. What functions are recognized by the vitamins investigated? Please add these references: vitamin E: "Dehbalaei, M. G., Ashtary-Larky, D., Mesrkanlou, H. A., Talebi, S., & Asbaghi, O. (2021). The effects of magnesium and vitamin E co-supplementation on some cardiovascular risk factors: A meta-analysis. Clinical nutrition ESPEN, 41, 110-117",Taheri, S., Asadi, S., Nilashi, M., Abumalloh, R. A., Ghabban, N. M., Yusuf, S. Y. M., ... & Samad, S. (2021). A literature review on beneficial role of vitamins and trace elements: evidence from published clinical studies. Journal of Trace Elements in Medicine and Biology, 67, 126789", vitamin A: "Sinopoli, A., Caminada, S., Isonne, C., Santoro, M. M., & Baccolini, V. (2022). What are the effects of vitamin A oral supplementation in the prevention and management of viral infections? A systematic review of randomized clinical trials. Nutrients, 14(19), 4081", VITAMIN C: "Juneja, D., Gupta, A., Kataria, S., & Singh, O. (2022). Role of high dose vitamin C in management of hospitalised COVID-19 patients: A minireview. World Journal of Virology, 11(5), 300".
I think that lacking an effective reflection in the discussion. It is important to stimulate the reader to future reflections.
Author Response
Thanks for your review. Your comments help us and encourage us to improve our research. We request your questions point by point below. The corrections suggested by you and other reviewers are highlighted in the original manuscript in red print.
Thank you for the opportunity to review this manuscript.
The ubiquitous exposure to vitamins has led to many epidemiological studies for many diseases. Glaucoma is no exception. Several studies suggested that nutrition might affect intraocular pressure or glaucoma mediated by oxidative stress. Oxidative stress occurs when more reactive oxygen species are formed than the anti-oxidative capacity of the cell can handle.
This study aims to investigate whether high intakes of some antioxidant vitamins, such as vitamins A, C, and E, have a protective effect on glaucoma.
I have some suggestions to improve this manuscript. In particular, the introduction should better investigate the role of vitamins analyzed in clinical management and proposal in this manuscript. What functions are recognized by the vitamins investigated? Please add these references: vitamin E: "Dehbalaei, M. G., Ashtary-Larky, D., Mesrkanlou, H. A., Talebi, S., & Asbaghi, O. (2021). The effects of magnesium and vitamin E co-supplementation on some cardiovascular risk factors: A meta-analysis. Clinical nutrition ESPEN, 41, 110-117",Taheri, S., Asadi, S., Nilashi, M., Abumalloh, R. A., Ghabban, N. M., Yusuf, S. Y. M., ... & Samad, S. (2021). A literature review on beneficial role of vitamins and trace elements: evidence from published clinical studies. Journal of Trace Elements in Medicine and Biology, 67, 126789", vitamin A: "Sinopoli, A., Caminada, S., Isonne, C., Santoro, M. M., & Baccolini, V. (2022). What are the effects of vitamin A oral supplementation in the prevention and management of viral infections? A systematic review of randomized clinical trials. Nutrients, 14(19), 4081", VITAMIN C: "Juneja, D., Gupta, A., Kataria, S., & Singh, O. (2022). Role of high dose vitamin C in management of hospitalised COVID-19 patients: A minireview. World Journal of Virology, 11(5), 300".
I think that lacking an effective reflection in the discussion. It is important to stimulate the reader to future reflections.
Thanks a lot for your kind comments. Following the suggestions of the four reviewers we have performed new analysis, included new tables and improved the results and the discussion sections. We have also read your articles, and now include them in the new version of the manuscript. Thanks, they were very useful. We have also followed your advice about the discussion, and we now hope it will be able to stimulate the reader to future reflections as you suggested. We believe the effect of vitamin intake on different pathologies is a very interesting study area and deserves furthers studies. Thanks a lot for your comments, and I hope you find the new version improved.
Round 2
Reviewer 2 Report
Thank you for these modifications, I have no further comments.